# Mechanical and Physical Properties of Particleboard Made from the Sumatran Elephant (*Elephas maximus sumatranus*) Dung and Wood Shaving

**DOI:** 10.3390/polym14112237

**Published:** 2022-05-31

**Authors:** Rudi Hartono, Ahmad Mubarok Dalimunthe, Apri Heri Iswanto, Evalina Herawati, Jajang Sutiawan, Afonso R. G. de Azevedo

**Affiliations:** 1Department of Forest Products, Faculty of Forestry, Universitas Sumatera Utara, Medan 20155, Indonesia; ahmadmubarok1998@gmail.com (A.M.D.); apri@usu.ac.id (A.H.I.); evalina@usu.ac.id (E.H.); jajangsutiawan@usu.ac.id (J.S.); 2LECIV—Civil Engineering Laboratory, UENF—State University of the Northern Rio de Janeiro, Av. Alberto Lamego, 2000, Campos dos Goytacazes 28013-602, RJ, Brazil; afonso.garcez91@gmail.com

**Keywords:** elephant dung fibers, isocyanate, particleboard, wood shavings

## Abstract

Sumatran elephants (*Elephas maximus sumatranus*) are the world’s largest living land mammals. The elephant’s digestive system can only absorb 40% of the nutrients in digested feed, and the remainder is excreted as dung. Elephant dung waste can be used as a particleboard material due to its high fiber content. The objectives of this study are: (i) to prepare elephant dung waste as raw material for particleboard, (ii) to improve elephant dung particleboard’s physical and mechanical properties using wood shavings, and (iii) to study the influence of several parameters on the physical and mechanical properties of particleboard. The particleboard dimensions and density were set at 20 cm × 20 cm × 1 cm and 0.8 g/cm^3^, respectively. The mixture ratio of elephant dung and wood shavings was 100/0, 90/10, 80/20, 70/30, 60/40, and 50/50 (% *w/w*). This mixture ratio of particles was sprayed with 7% isocyanate adhesive. The pressing at a pressure of 30 kg/cm^2^ for 5 min and 160 °C was used in this study. The physical and mechanical properties of particleboard were tested according to JIS A 5908 (2003) standard. The result shows that the addition of wood shaving improved the elephant dung particleboard’s physical and mechanical properties. Except for moisture content and water absorption, the addition of wood shavings has a significant effect on elephant dung particleboard’s physical and mechanical properties. The best ratio of elephant dung and wood-shaving for this research is 50/50 and has fulfilled the JIS A 5908-2003 standard, except for thickness swelling.

## 1. Introduction

Production of wood-based panels experienced a decline in 2020 due to the impact of the COVID-19 pandemic. The decline in production was 3.3% overall. In addition, consumption of wood-based panels also experienced a decline in 2020, around 4.3% [1]. Consumption of structural panels fell 2.2%, while consumption of non-structural panels fell 5.6% [1]. Since the COVID-19 pandemic has begun to end in 2021, FAO is optimistic that there will be an increase in the production of wood-based panels [1]. The potential for this increase will certainly increase the raw materials for making wood-based panels such as raw material for particleboard.

Utilized natural fibers or particles as an alternative raw material are important for particleboard production due to the decline of forest areas [2]. Many natural fibers have been utilized as raw materials for fabricated particleboard. Those materials include oil palm trunk [3], rice husk [4], sugar cane bagasse [5], corn stalk [6], nipah frond [7], salacca frond [8], kenaf [9], and sorghum [10]. Reducing environmental challenges and impacting ecological and socio-economic aspects of resource management are several advantages of utilizing a natural fiber [11]. In addition, Azevedo et al. [12] reported that application in the development of alternative composite materials is one way to manage this natural fiber or agro-industrial waste effectively.

Sumatran elephant dung waste (*Elephas maximus sumatranus*) can be used as a particleboard material due to its high fiber content. The Sumatran elephant is a ruminant animal that can only absorb 40% of nutrients [13]. Sumatran elephants produce up to 100–130 kg of dung per day and are still widely used as biogas producers [14]. Farah et al. [15] successfully used elephant dung waste to manufacture exotic paper. In addition, a previous study successfully utilized elephant dung as materials particleboard [16,17].

Jati et al. [16] reported that particleboard elephant dung fabricated at 0.8 kg/cm^3^ in density with 10% citric acid has the best composition. The modulus of elasticity and modulus of rupture of particleboard satisfied the standard JIS A 5908-2003. However, fabricated particleboard’s dimensional stability and internal bonding do not satisfy JIS A 5908-2003 standard. Widyorini et al. [17] optimized the pressing temperature and citric acid content to overcome these situations. The study reported that the dimensional stability of particleboard from elephant dung fibers improved significantly with increasing pressing temperature and citric acid content. The pressing temperature of 200 °C and 20 wt% of citric acid content is optimal in this research.

Another method to potentially improve particleboard’s physical and mechanical properties from elephant dung is the addition of wood particles [18,19,20]. Iswanto et al. [21] investigated the ratio of jatropha fruit hulls and wood shavings in particleboard. The increasing proportion of wood shavings increased the physical and mechanical properties. Iswanto et al. [22] reported that wood shaving successfully improves sorghum bagasse particleboard’s dimensional stability. In addition, Guller et al. [23] reported that the particleboard from sunflower stalks and Calabrian pine sawdust was found at a 50/50 ratio. Therefore, the objectives of this study are: (i) to prepare elephant dung waste as raw material for particleboard, (ii) to improve elephant dung particleboard’s physical and mechanical properties using wood shavings, and (iii) to study the influence of several parameters on the physical and mechanical properties of particleboard.

## 2. Materials and Methods

### 2.1. Materials

Elephant dung is collected at the Aek Nauli Conservation Camp in Simalungun Regency (Medan, Indonesia), while mahogany wood shavings (*Swietenia mahagoni*) waste is collected in Wood Working Industry Medan (Medan, Indonesia). The elephant dung fibers and wood shavings are shown in Figure 1.

First, the elephant dung was washed and dried in the sun. Secondly, the elephant dung particles and wood shavings materials were dried in an oven at 100 °C until they had a moisture content of 8%. The dimension of a particle in this study was calculated using a digital caliper. In addition, mass water absorption of a particle was calculated before and after soaking for 24 h. Table 1 shows the dimensions, moister content and water absorption capacity of elephant dung fiber and wood shavings. 

### 2.2. Particleboard Production and Testing 

The particleboard dimensions and density were set at 20 cm × 20 cm × 1 cm and 0.8 g/cm^3^, respectively. First, the mixture ratio of elephant dung and wood shavings was 100/0, 90/10, 80/20, 70/30, 60/40, and 50/50 (% *w/w*) (Figure 2). This mixture of particles was sprayed with 7% isocyanate adhesive (Table 2). Second, the mixture is then placed in a mat-forming box without opening time. Third, pressing at a pressure of 30 kg/cm^2^ for 5 min and 160 °C was used in this study. Finally, the particleboard was conditioned for approximately seven days at a room temperature of 20 °C and a relative humidity of approximately 60%. After conditioning, particleboard was cut and tested according to JIS A 5908 (2003) standard [24]. The tests conducted on particleboard include those on its physical properties, such as density, moisture content (MC), thickness swelling (TS), and water absorption (WA), as well as its mechanical properties, such as bending properties, i.e., the modulus of elasticity (MOE), modulus of rupture (MOR), and internal bonding (IB). Each experiment was performed in three replications, and the average values and standard deviations were calculated.

Density testing was performed using samples measuring length, width, and thickness (5 × 5 × 1) cm^3^. The density determination was expressed in the comparison results between the mass and volume of the board. The MC test was carried out using a sample measurement (5 × 5 × 1) cm^3^. The MC test was calculated based on the initial and final mass after drying in the oven for 24 h at 103 ± 2 °C. The WA test was carried out using a sample measuring (5 × 5 × 1) cm^3^. The WA test was obtained from the difference between the mass before and after soaking for 24 h. The TS test was carried out using a sample measuring (5 × 5 × 1) cm^3^. The TS is obtained from the difference between the initial thickness before soaking and the thickness after 24 h.

The MOE and MOR tests were carried out using samples measuring length, width, and thickness (20 × 5 × 1) cm^3^. Static bending one-point loading of MOE and MOR of the samples was also evaluated. The test was carried out using a Universal Testing Machine with a 10 mm/minute loading speed. The IB test used samples measuring length, width, and thickness (5 × 5 × 1) cm^3^. The sample was glued to two iron blocks with epoxy glue and allowed to dry for 2 × 24 h. The two iron blocks were then pulled perpendicular to the sample’s surface to the maximum load using a Universal Testing Machine with a loading speed of 2 mm/min.

### 2.3. Morphology Analysis

According to Sutiawan et al. [25], particleboard’s surface morphology was analyzed using Digital Microscope (Dino-Lite) analysis. The particle distribution in particleboard was analyzed with a magnification of 50×.

### 2.4. Data Analysis

A non-factorial, completely randomized design (CRD) with one treatment factor (adding wood shaving) and three replications were used in this study. Duncan was conducted (DMRT) if the treatment had a significant effect at a *p*-value *<* 0.05.

## 3. Results and Discussions

### 3.1. Physical Properties

The particleboard’s density and MC is illustrated in Figure 3. The density of particleboard varies between 0.63 and 0.68 g/cm^3^. The highest particleboard density is found at a 50/50 ratio (0.68 g/cm^3^), and the lowest is found at a 100/0 ratio (0.63 g/cm^3^). The data analysis revealed that the addition of wood shaving significantly affected particleboard density (Table 3). This result indicates that increasing the proportion of wood shavings statically increases the board’s density (Table 4). These phenomena are due to the difference in the raw materials’ density, with the elephant dung material having a low density. According to Widyorini et al. [17], elephant dung has a bulk density of 0.11 ± 0.002 g/cm^3^. The addition of wood shavings of Mahoni with a density of 0.60 ± 0.08 g/cm^3^ increases the density of elephant dung particleboard. Gilbero et al. [26] reported that Mahoni density is around 0.58–0.63. 

The density particleboard in this study does not meet the target of 0.8 g/cm^3^. These phenomena are due to the board’s thickness being increased during conditioning (spring back). The spring back value for this study is between 3.43 and 12.63% (Table 5). The highest spring back value for particleboard is found at a 100/0 ratio (12.63%), and the lowest spring back value is found at a 50/50 ratio (3.43%). The spring back phenomenon is found on the particleboard during conditioning. Adjusting the moisture content increases the particleboard’s thickness and eventually decreases its density [27,28,29,30,31]. The density of particleboard in this study is greater than particleboard mixed with rattan shavings and sawdust (0.57–0.60 g/cm^3^) [32] and particleboard mixed with coconut powder and sengon wood (0.54–0.58 g/cm^3^) [33]. According to JIS A 5908-2003, all particleboard densities obtained were within the standard range (0.4–0.9 g/cm^3^). 

The MC of particleboard ranges between 8.36 and 9.78%. The highest MC of particleboard is found at a ratio of 100/0 (9.78%). Meanwhile, the lowest MC of particleboard is found at a ratio of 60/40 (8.36%). The data analysis revealed that the addition of wood shaving to the MC was statistically significant (Table 3). According to JIS A 5908-2003 standard, the MC of the particleboard obtained met the standard (5–13%). 

The WA and TS of particleboard as shown in Figure 4. WA in this study ranged between 58.32% and 67.74%. The highest value was 67.74% at a ratio of 100/0, and the lowest value was 58.32% at a ratio of 50/50. The data analysis indicates that the effect of the addition of wood shaving on the WA of the board is not statistically significant (Table 3). The study discovered that increasing the proportion of wood shavings resulted in decreased WA. Due to the low density of elephant dung particles, they have a higher water absorption capacity of 372.21 ± 18.65% than wood shavings of 223.05 ± 16.19% (Table 1). In addition, the particle sizes of the raw material wood shavings have larger dimensions than elephant dung fiber. Smaller particle size results in a larger surface area/contact area, which affects the particleboard’s physical and mechanical properties. According to Pan et al. [34], fine particles provided a bigger contact area of the particleboard than coarse particles.

The TS has a value between 20.69% and 36.5%, with the highest value being 36.5% in the 100/0 ratio and the lowest value being 20.69% in the 50/50 ratio. The data analysis demonstrates that the addition of wood shaving significantly affected the TS of particleboard (Table 3). It is demonstrated in this study that increasing the proportion of wood shavings results in a statistically significant decrease in the TS value obtained (Table 4). These phenomena are due to the addition of wood shavings with higher specific gravity than elephant dung fiber. According to the JIS A 5908-2003 standard, the thickness swelling value of all particleboard in this study does not meet the standard, which specifies a maximum thickness swelling of 12%.

### 3.2. Mechanical Properties

The MOE and MOR, as illustrated in Figure 5. The MOE in this study ranges between 1952 MPa and 2573 MPa. The highest MOE of particleboard is 2573 MPa at a 50/50 ratio, while the lowest is 1952 MPa at a 100/0 ratio. The higher the wood shaving ratio, the greater the MOE produced in this study (Table 6 and Table 7). These phenomena are due to wood shavings particles of larger dimensions than elephant dung fibers (Table 1). Lias et al. [35] reported that using coarse particles would increase the MOE rather than a fine particle. According to JIS A 5908-2003, except for the 100/0 ratio, all particleboard fabricated MOE values met the standard. 

The particleboard’s MOR is 18.6–27.4 MPa. The highest MOR values is found on the 50/50 ratio, while the lowest MOR values is found on the 90/10 ratio. The data analysis revealed that the addition of wood shaving has significantly affected the MOR of particleboard (Table 6). The increase in MOR value at a 50/50 ratio is due to coarse wood shavings and large dimensions at a similar MOE. This study was similar to Guller et al. [23], which discovered mechanical properties in the highest MOR at a 50/50 ratio of sunflower stalks to Calabrian pine. According to JIS A 5908-2003 standard, the MOR value of all particleboard produced met the standard, which requires a minimum MOR value of 8 MPa. 

As illustrated in Figure 6, particleboard’s IB varies between 0.16–0.34 MPa, with the highest value occurring at the 50/50 ratio and the lowest occurring at the 100/0 ratio. It was discovered in this study that there was a tendency to increase the ratio of wood shavings, resulting in a higher internal bond value (Table 7). These phenomena were due to the board density 50/50 ratio being larger than the 100/0 ratio. Umemura et al. [36] reported that IB increase with the increasing density of particleboard. According to the JIS A 5908-2003 standard, all particle boards have an IB value greater than 0.15 MPa. 

### 3.3. Morphology of Particleboard

The surface morphology of the particleboard in this study is demonstrated in Figure 7. Figure 7a describes the small homogenous particle of elephant dung. Meanwhile, Figure 7b describes the combination of small particle elephant dung and particleboard coarse particle wood shaving. Small particles could fill the void space in the coarser particles. Therefore, the excellent mechanical properties of particleboard, such as MOE and MOR, were found [37].

## 4. Conclusions

The addition of wood shaving improved the physical and mechanical properties of the elephant dung particleboard. Except for moisture content and water absorption, the addition of wood shavings has a significant effect on elephant dung particleboard’s physical and mechanical properties. The best ratio of elephant dung and wood-shaving for this research is 50/50 and has fulfilled the JIS A 5908-2003 standard, except for thickness swelling. The combination of small particle elephant dung and particleboard coarse particle wood shaving at a 50/50 ratio resulted in the excellent modulus of elasticity and rupture of particleboard.

## Figures and Tables

**Figure 1 polymers-14-02237-f001:**
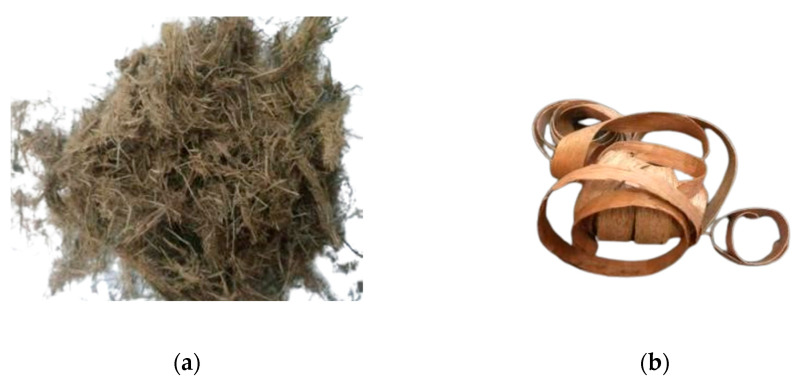
(**a**) The elephant dung fibers and (**b**) wood shavings in this study.

**Figure 2 polymers-14-02237-f002:**
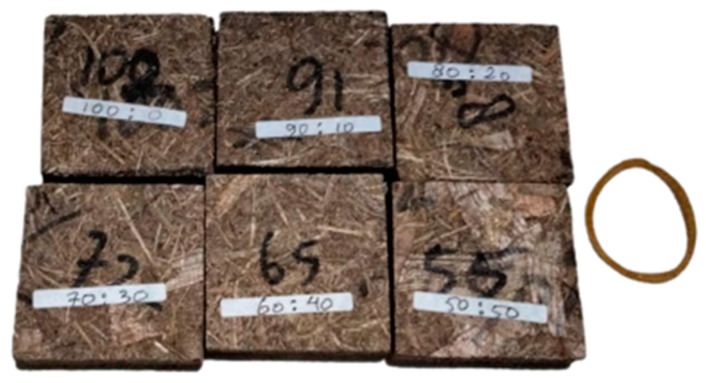
Photograph elephant dung fibers and wood shavings particleboard in this study.

**Figure 3 polymers-14-02237-f003:**
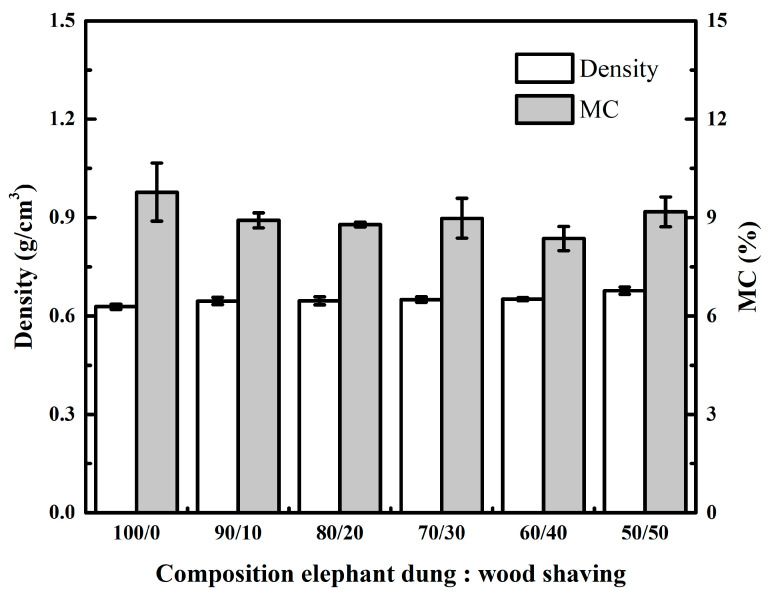
Effect of wood shaving on density and MC of elephant dung particleboard.

**Figure 4 polymers-14-02237-f004:**
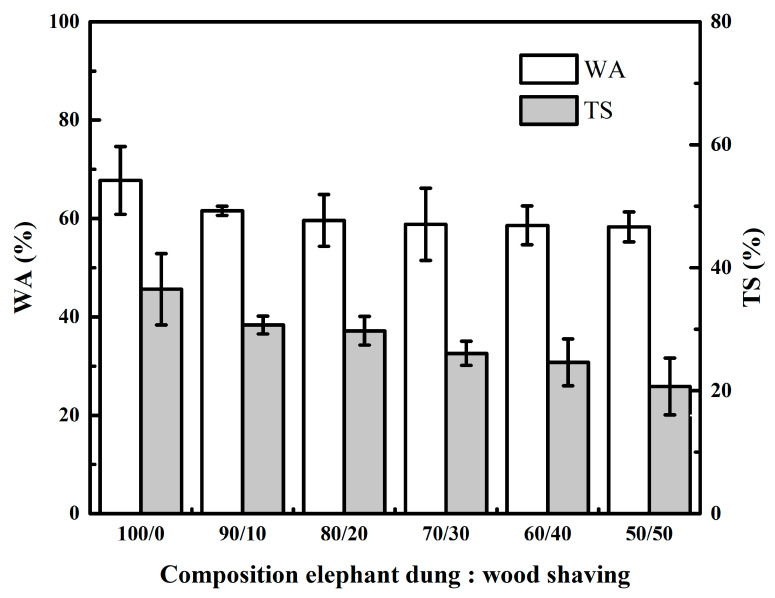
Effect of wood shaving on WA and TS of elephant dung particleboard.

**Figure 5 polymers-14-02237-f005:**
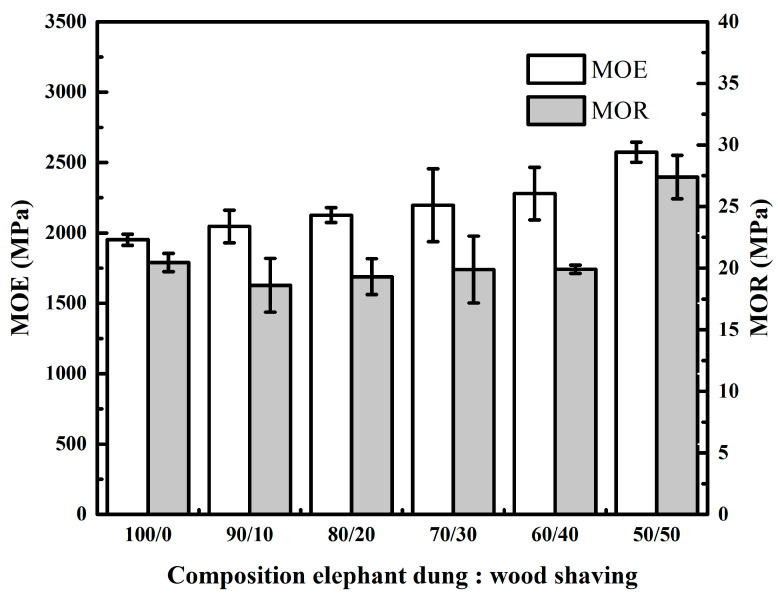
Effect of wood shaving on MOE and MOR of elephant dung particleboard.

**Figure 6 polymers-14-02237-f006:**
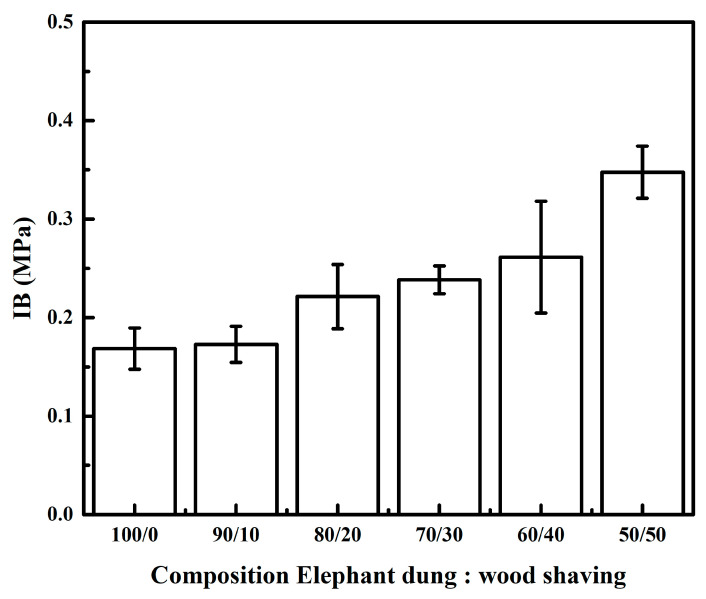
Effect of wood shaving on the IB of elephant dung particleboard.

**Figure 7 polymers-14-02237-f007:**
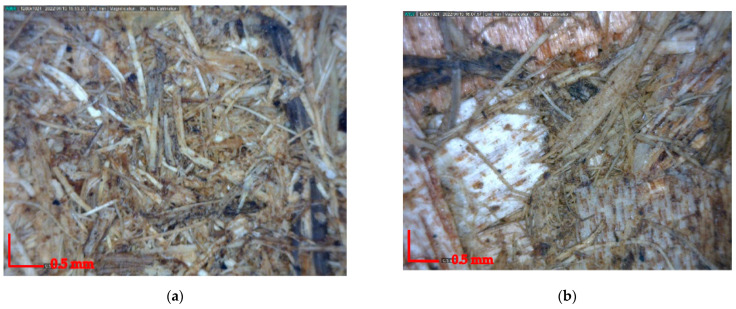
Digital Microscope analysis (**a**) 100/0 and (**b**) 50/50 the mixture ratio of elephant dung and wood shavings at magnification 50×.

**Table 1 polymers-14-02237-t001:** Size, moisture content, and water absorption capacity of wood shavings and elephant dung.

Parameters	Wood Shaving	Elephant Dung
Length (mm)	540 ± 336.56	37.31 ± 17.23
Width (mm)	15.84 ± 1.96	0.51 ± 0.24
Thickness (mm)	0.59 ± 0.22	0.26 ± 0.13
Moisture content (%)	8%	8%
Water absorption (%)	223.05 ± 16.19	372.21 ± 18.65

**Table 2 polymers-14-02237-t002:** The properties of isocyanate adhesive in this study.

Properties	Information
Solids content (%)	98
Viscosity (Cps/23 °C)	150–250
pH	6.5–8.5
Pot life mixture	Within 60 min
Spread volume	200~250 g/m^2^
Assembly time	Within 10 min
Cold Press	Low density wood 7–10 kg/cm^2^ High density wood 10–15 kg/cm^2^
Press time	Cold Press 30–45 min depends on wood species, size of lamella, temperature and spread volume.

Source: PT. Polychemie Asia Pasific (Jakarta, Indonesia).

**Table 3 polymers-14-02237-t003:** Variance analysis summary of physical properties.

Parameter	ANOVA
Density	0.002 **
MC	0.001 **
WA	0.080 ^ns^
TS	0.004 **

^ns^ not significance, ** highly significance difference.

**Table 4 polymers-14-02237-t004:** Duncan multi-range-test of physical properties.

Composition (% *w/w*)	Density	MC	WA	TS
100/0	0.63 a	9.78 cd	67.74 b	36.50 c
90/10	0.65 b	7.93 a	61.57 ab	27.94 b
80/20	0.65 b	8.79 ab	53.34 a	29.75 b
70/30	0.64 b	10.32 d	58.83 ab	26.09 ab
60/40	0.65 b	8.36 ab	58.62 ab	24.61 ab
50/50	0.68 c	9.18 bc	58.32 ab	20.69 a

Value with the same letter within a row is not significantly different.

**Table 5 polymers-14-02237-t005:** Particleboard spring back value.

Composition (% *w/w*)	Spring Back (%)
100/0	12.63
90/10	8.70
80/20	8.53
70/30	7.57
60/40	6.90
50/50	3.43

**Table 6 polymers-14-02237-t006:** Variance analysis summary of mechanical properties.

Parameter	ANOVA
MOE	0.003 **
MOR	0.000 **
IB	0.000 **

^ns^ not significance, ** highly significance difference.

**Table 7 polymers-14-02237-t007:** Duncan multi-range-test of mechanical properties.

Composition (% *w/w*)	MOE	MOR	IB
100/0	1952 a	20.4 a	1.69 a
90/10	2046 ab	18.6 a	1.73 a
80/20	2127 ab	19.3 a	2.21 ab
70/30	2197 ab	20.2 a	2.38 b
60/40	2280 b	19.9 a	2.61 b
50/50	2573 b	27.4 b	3.48 c

Value with the same letter within a row is not significantly different.

## Data Availability

The data presented in this study are available on request from the corresponding author.

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
