# Peer review of "Mechanical and Physical Properties of Particleboard Made from the Sumatran Elephant (Elephas maximus sumatranus) Dung and Wood Shaving"

_polymers, 2022, doi:10.3390/polym14112237_

Round 1

Reviewer 1 Report

Manuscript No.: polymers-1724733-peer-review-v1

Title: Improving Properties of Particleboard Made from the Sumatran Elephant (Elephas maximus sumatranus) Dung Fiber using Wood Shaving

Authors: Hartono et. al.

Polymers

General

The article describes the influence of a mixture of elephant dung and wood particles on the mechanical properties of particleboard. The article is interesting and could be of practical value in certain parts of the world where the various materials, such as elephant dung in this case, could serve as material for making particleboard.

The article is generally well written and understandable, and the results are also understandable. Nevertheless, the article has some shortcomings, which are listed below.

Specific comments

L2-3 The title could be less complicated in one way and more descriptive in another way. Since the articled is investigating mechanical and physical properties, one possible tite could be:

Mechanical and physical Properties of Particleboard Made from the Sumatran Elephant (Elephas maximus sumatranus) Dung and Wood Shaving

L40-71 Write the purpose of the article more detailed.

L79 The first elephant dung was washed and dried in the sun. There is also a second dung? You probably meant: First, the elephant dung…

L85, Table1. The whole wood shaving (540 mm) long was used in the production or it was milled? In case the whole was used, how it was possible to stir the mixture, since dry wood shaving with dimension 540x15x0.6 mm are quite stiff??? This would be practically impossible! Explain this in more detail!

Show also some figures with mixture and figures of final boards!

L89 Describe the adhesive more detailed. What was the opening times, etc? What was the time between applying the glue and pressing?

L98-115 How many specimens were used in each properties determination?

L92 At what conditions were specimens conditioned?

L107-108 How was the MOE and MOR determined? Did you use bending test or some other test? Describe more detailed!

L149 What would be the MC for dung alone (not as particleboard)?

L162-164 In addition, the particle sizes of the raw material wood shavings have larger than elephant dung fiber. Larger what?

L165-167 According to Pan et al. [34], the surface area/contact area of fine particles (40-60 mesh) is too large for adhesives with the same adhesive ratio to penetrate (20-40 mesh). This sentence is not clear.

L220, Figure 6 The scale is not clear.

Considering all results: When explaining the significant difference using p value, you are only mentioning if there is significant difference or not only in general. For example in L195-196:

The data analysis revealed that the addition of wood shaving has significantly affected the modulus of rupture of particleboard (p<0.05).

I am sure there is a significant difference between a 60/40 and a 50/50. But I am also sure that there is no significant difference between 90/10 and 80/20. But you are generally saying as if there is a difference between all values. Make additional tables with the p-values between each composition of elephant dung and wood shaving. Do this for all results so the reader can see between which combinations there is a significant difference.

Author Response

Dear Reviewer,

Thank you very much for your comment, suggestion, and recommendation in our manuscript. All changes in our manuscript we highlight in yellow.

Best Regard

Rudi Hartono

Reviewer 2 Report

Comment for polymers-1724733 is listed as follows,

  1. There are some miss been named or error typing in the pdf file of manuscripts.

(1) In the section Keywords: please change the "isocyanates" into the "isocyanate".

(2) In the subsection 3.1: please change the "moisture content (MC)" into the "MC", please change the "thickness swelling (TS)" into the "TS", please change the "water absorption (WA)" into the "WA".

(3) In the subsections 3.2 and 3.3: please change the "modulus of elasticity (MOE)" into the "MOE", please change the "modulus of rupture (MOR)" into the "MOR", please change the "internal bond (IB) " into the "IB".

(4) In Figure 4, please check the unit "MPa".

Author Response

(The authors gave the same response as above.)

Round 2

Reviewer 1 Report

Manuscript No.: polymers-1724733-peer-review-v1

Title: Improving Properties of Particleboard Made from the Sumatran Elephant (Elephas maximus sumatranus) Dung Fiber using  Wood Shaving

Authors: Hartono et. al.

Polymers

 General

The authors have significantly corrected the article according to the comments. Nevertheless, they have not considered all the comments:

L40-71 Write the purpose of the article more detailed.

The purpose should be written more extensively, not adding just one line.

L89 Describe the adhesive more detailed. What was the opening times, etc? What was the time between applying the glue and pressing?

The properties of the glue have not been described more detailed.

L107-108 How was the MOE and MOR determined? Did you use bending test or some other test? Describe more detailed!

In the text you have not added that MOE and MOR were determined with bending test (as you mentioned in reply). Describe this test in more detail (3 or 4 point bending test,…)

L149 What would be the MC for dung alone (not as particleboard)?

Your reply: Thank you very much for your comment. The elephant dung particles materials were dried in an oven at 100 °C until had a moisture content of 8%

First: The data you are mentioning in the reply I have not found in the article.

And second: Beginning with line 149 (original article) you are explaining the moisture contents of the boards, which were conditioned at 20°C and 60% relative air humidity and not dried in oven. So you should add data of moisture content of dung alone conditioned at 20°C and 60%.

L220, Figure 6 The scale is not clear.

The scale is still missing.

Author Response

Dear Reviewer,

Thank you very much for your comment, suggestion, and recommendation. All changes in our manuscript we highlight in yellow.

Best Regard

Rudi Hartono
